# Current and Emerging Therapeutic Targets for the Treatment of Cholangiocarcinoma: An Updated Review

**DOI:** 10.3390/ijms25010543

**Published:** 2023-12-30

**Authors:** Matthew J. Hadfield, Kathryn DeCarli, Kinan Bash, Grace Sun, Khaldoun Almhanna

**Affiliations:** 1Division of Hematology/Oncology, Department of Medicine, The Warren Alpert School of Medicine of Brown University, Providence, RI 02806, USA; matthew_hadfield@brown.edu (M.J.H.); grace_sun@brown.edu (G.S.); 2Department of Graduate Studies, University of New England, Biddeford, ME 04005, USA; kinanbash@gmail.com

**Keywords:** hepatobiliary malignancy, gallbladder cancer, cholangiocarcinoma, biomarker, targeted therapy

## Abstract

Cholangiocarcinoma is a malignancy of the bile ducts that is often associated with late diagnosis, poor overall survival, and limited treatment options. The standard of care therapy for cholangiocarcinoma has been cytotoxic chemotherapy with modest improvements in overall survival with the addition of immune checkpoint inhibitors. The discovery of actionable mutations has led to the advent of targeted therapies against FGFR and IDH-1, which has expanded the treatment landscape for this patient population. Significant efforts have been made in the pre-clinical space to explore novel immunotherapeutic approaches, as well as antibody–drug conjugates. This review provides an overview of the current landscape of treatment options, as well as promising future therapeutic targets.

## 1. Introduction

### 1.1. Cholangiocarcinoma Epidemiology

Cholangiocarcinoma, predominantly adenocarcinomas, encompasses gallbladder cancers, ampullary carcinomas, and intrahepatic and extrahepatic cholangiocarcinoma. Tumors of the biliary tract are usually classified based on their anatomical origin, intra- or extrahepatic. While these cancers are distinct and heterogenous, they are often studied together due to low incidence, with approximately 12,000 cases occurring in the United States per year [1]. Gallbladder cancer is more common among women and patients with prior cholelithiasis, calcification of the gallbladder wall, pancreaticobiliary maljunction, or gallbladder polyps. Additionally, chronic inflammation is a common risk factor for both gallbladder cancer and cholangiocarcinoma, which is associated with primary sclerosing cholangitis, cirrhosis, infectious hepatitis, and recurrent bile duct calculi. In East Asia, chronic inflammation from liver fluke infections contributes to an increased incidence of high-risk disease [2]. These malignancies are often diagnosed at a late stage. Prognosis is poor, with median survival in gallbladder cancer ranging from 5.8 months for metastatic disease to 12.9 months for earlier-stage disease [3]. The 5-year survival rate for resectable cholangiocarcinoma ranges from 16% to 52%, with a high risk of disease recurrence [4].

### 1.2. Standard Treatment

Only about one-third of patients with cholangiocarcinoma and malignancies of the gallbladder present with localized disease [2]. The standard curative treatment for early-stage disease is surgical resection, with or without neoadjuvant platinum doublet or fluoropyrimidine-based chemotherapy. While the programmed death ligand-1 (PD-1) inhibitor durvalumab was recently approved for the treatment of metastatic disease, its role has yet to be established in the neoadjuvant setting. Select patients with unresectable disease may be considered for liver transplant; however, the criteria for transplant in this patient population are not well established. The role of adjuvant chemotherapy has not been robustly studied due to low incidence, but there are data to support prolonged survival in patients who receive adjuvant chemotherapy or chemoradiation [5,6,7,8].

Most patients present with unresectable or metastatic disease and may be offered clinical trials, best supportive care, or systemic therapy. Gemcitabine plus cisplatin has been considered the standard of care first-line systemic therapy since 2010, based on the results of the phase III ABC-02 study [9]. In 2022, the phase III TOPAZ-1 trial demonstrated survival benefit with the addition of durvalumab to this regimen, leading to the adoption of concurrent chemoimmunotherapy as the new standard of care [10]. Additional accepted systemic therapies may include fluoropyrimidine-based chemoradiation, other gemcitabine-based chemotherapy regimens, or targeted treatments selected based on tumor characteristics and molecular profiling. Unfortunately, the overall survival rate for metastatic disease remains less than one year, highlighting a need for advances in targeted therapy and additional therapeutic combinations to improve survival. Frequently, the treatment of biliary tumors can be complicated by biliary obstruction and liver dysfunction. These complications limit treatment options, making this a challenging patient population to treat.

## 2. Molecular Targets

### 2.1. Overview

Based on a recently published genomic analysis, cholangiocarcinoma has proven to be a very heterogenous disease depending on tumor location [11]. This has led to the identification of genomic patterns that correlate with both disease prognosis and potential therapeutic targets (Figure 1). Over 30% of biliary tract malignancies, including cholangiocarcinoma, harbor an actionable mutation with druggable targets [12]. Targeted agents have played an increasing role in the treatment of multiple solid tumor malignancies with the discovery of more actionable mutations, and the utilization of these agents is evolving in the treatment of cholangiocarcinoma. Molecular testing of all unresectable cholangiocarcinoma is now the recommended standard of care to inform treatment decisions and determine eligibility for clinical trial enrollment. In the last several years, several new agents have demonstrated clinical activity in the treatment of cholangiocarcinoma. The benefits, however, have been modest. In this article, we review the most common targetable molecular alterations that are now part of the standard of care for treatment. We also cover the evolving treatment landscape for biliary tract cancers, including novel immune checkpoint inhibitors, adoptive cellular therapies, and drugs targeting the tumor microenvironment and cellular metabolism and the role they will play in the future treatment of hepatobiliary malignancies. Table 1 summarizes recent and ongoing trials of targeted agents in hepatobiliary cancers.

### 2.2. BRAF V600E

B-Raf proto-oncogene, serine/threonine kinase (BRAF), is a serine/threonine kinase signaling pathway that is one of the most mutated pathways in solid tumors, including biliary cancers. BRAF V600E mutations are seen in approximately 5% of biliary cancers [13]. Treatment response with single-agent BRAF inhibition is often limited due to acquired resistance mutations in the MAPK pathway. The combination of BRAF and MEK inhibitors has been shown to provide more durable responses and received tissue-agnostic FDA approval in 2022 [14]. The combination therapy of the BRAF inhibitor, dabrafenib, and the MEK inhibitor, trametinib, has been studied in BRAF V600E-mutated cholangiocarcinoma. This regimen showed clinical activity among patients with metastatic BRAF V600E-mutated biliary tract cancers in a phase II clinical trial of 43 patients, with an ORR of 46.5% [13]. Dabrafenib plus trametinib combination therapy is now FDA-approved as a subsequent line treatment option for this patient population. Several ongoing trials are now testing different combinations of the above drugs in patients with BRAF mutations, including cholangiocarcinoma.

### 2.3. EGFR

Epidermal growth factor receptor (EGFR) is the binding membrane receptor for the epidermal growth factor family of proteins. These proteins play a role in regulating cellular proliferation, angiogenesis, and apoptosis [15]. Activating EGFR mutations occur across solid tumors and can be present in up to 15% of hepatobiliary malignancies [2]. The utilization of EGFR-targeting tyrosine kinase inhibitors (TKIs) has been evaluated in small phase II studies of patients with cholangiocarcinoma. Panitumumab, an anti-EGFR monoclonal antibody, demonstrated anti-tumor activity and tolerable safety profile in combination with gemcitabine and irinotecan in a phase II trial of patients with unresectable cholangiocarcinoma [16]. An additional phase II study evaluated the EGFR targeting TKI, Erlotinib, in 42 patients with EGFR-mutated advanced biliary cancer. The study demonstrated that 17% of patients were progression-free at 6 months of 36 evaluable patients [17]. Despite initial promising results from early studies of cetuximab and erlotinib, these have not been born out in subsequent trials [18,19]. The clinical benefit of targeting EGFR in biliary tract malignancies has been less significant than other solid tumors with exceptional responses to these TKIs, such as EGFR-mutated non-small cell lung cancer. The difference in responses between these tumor types highlights the heterogeneity of responses to targeted therapy and warrants further investigation.

### 2.4. FGFR

Fibroblast growth factor receptor (FGFR) is a group of tyrosine kinase receptors that have a functional role in cellular proliferation and growth. Mutations within the FGFR family of proteins have been implicated in the pathogenesis of cholangiocarcinoma [20]. FGFR-2 mutations are seen in approximately 14% of intrahepatic cholangiocarcinoma and are more common in intrahepatic than extrahepatic cancers [21]. They are associated with increased intraductal cell proliferation and tumorigenesis [2]. Clinically, FGFR-2 mutations may confer more indolent disease and prolonged survival [22,23]. Tyrosine kinase inhibitors targeting FGFR have shown activity and tolerability in patients with FGFR2-mutated cholangiocarcinoma.

The FGFR1-3 inhibitor pemigatinib demonstrated anti-tumor activity in the single-arm phase II FIGHT-202 trial [24]. Among 107 patients with FGFR-2 mutated metastatic cholangiocarcinoma, the ORR was 35.5%, and disease control was achieved in 80% of patients. Toxicities of pemigatinib include electrolyte derangements, notably hyperphosphatemia, hypophosphatemia, and hyponatremia. Hyperphosphatemia was seen in 60% of patients, with 12% experiencing grade 3 AE. Additional toxicities included arthralgias, stomatitis, abdominal pain, fatigue, retinal detachment, and dry eye. The open-label randomized phase III FIGHT-302 trial is currently underway to study pemigatinib versus gemcitabine plus cisplatin chemotherapy as a first-line treatment option for metastatic cholangiocarcinoma [25].

An additional FGFR1-3 inhibitor, infigratinib, has also shown early activity in the treatment of FGFR2-mutated advanced cholangiocarcinoma. In a phase II trial of 108 patients with FGFR-2 gene fusions or rearrangements who were previously treated with at least one gemcitabine-containing regimen, the ORR was 23% [26]. Toxicities of infigratinib are similar to those of pemigatinib, including electrolyte derangements, stomatitis, dry eye, retinopathy, fatigue, and alopecia. Hyperphosphatemia was seen in 76% of patients, while severe hyponatremia and hypophosphatemia were each seen in 13%.

Most recently, the next-generation FGFR1-4 inhibitor futibatinib has also shown activity and a tolerable safety profile in the treatment of advanced cholangiocarcinoma with FGFR-2 fusions or rearrangements. The open-label phase II FOENIX-CCA2 trial included 103 patients with previously treated disease and demonstrated an ORR of 42% [27]. Grade 3 hyperphosphatemia occurred in 30% of patients with less frequent AEs, including increased aspartate aminotransferase level, stomatitis, and fatigue.

These three oral agents are all FDA-approved as subsequent line treatment options in the treatment of advanced FGFR-2 mutated cholangiocarcinoma after disease progression. They offer similar safety profiles and promising clinical applications. In the above trials, futibatinib was given once daily on a continuous dosing schedule, pemigatinib was given once daily for 14 days of a 21-day cycle, and infigratinib was given once daily for 21 days of a 28-day cycle. These options are convenient for patients who can reliably take medications at home and are now used in subsequent line standard treatment.

### 2.5. IDH1/IDH2

Isocitrate dehydrogenase (IDH) is an important enzyme for cellular metabolism. Mutations in the genes encoding the metabolic enzymes isocitrate dehydrogenase IDH-1 and -2 are associated with developing premalignant biliary lesions [2]. IDH1/2 mutations are detected in 13–36% of intrahepatic cholangiocarcinoma and <1% of extrahepatic cholangiocarcinoma, with more women harboring IDH-1 mutations than men [2,28,29]. The IDH-1 inhibitor ivosidenib was shown to improve PFS versus placebo (2.7 vs. 1.4 months, *p* < 0.0001) among 124 patients with metastatic IDH1-mutated cholangiocarcinoma in a phase III trial. Grade 3 adverse events included hyperphosphatemia in 30% of patients, increased aspartate aminotransferase level in 7% of patients, and stomatitis and fatigue each in 6% of patients [24]. Ivosidenib is now FDA-approved as a subsequent line treatment option in this patient population after disease progression. Further clinical trials are currently underway with IDH-1/IDH-2 targeted therapies in patients with hepatobiliary tumors. Olutasidenib, an IDH-1 inhibitor, was previously studied in Acute Myeloid Leukemia and is currently being tested in a phase I/II clinical trial of patients with solid tumors with expansion cohorts, including hepatobiliary cancers [30]. Previous studies have shown that IDH-1/2 mutated cholangiocarcinoma displays increased evidence of DNA damage, making POLY-ADP Ribose Polymerase (PARP) inhibitors a potential therapeutic option [31]. A trial evaluating the PARPi, Olaparib, in patients with IDH-1/2-mutated glioblastoma and cholangiocarcinoma is currently ongoing.

### 2.6. Her-2/NEU/ERBB2/ERBB3

Human epidermal growth factor receptor 2 (HER-2) is a receptor tyrosine kinase receptor encoded by the ERBB2 gene. Application of the HER-2 gene leads to downstream upregulation of key receptor signaling pathways that contribute to cellular growth and proliferation [32]. HER-2 targeting agents have been deployed across multiple solid tumors, including breast and gastric, with significant clinical benefit and efficacy. Roughly 13% of gallbladder cancers, 18% of extrahepatic cholangiocarcinoma, and 5% of intrahepatic cholangiocarcinoma overexpress HER-2 via ERBB2 or ERBB3 mutations [33,34,35]. A retrospective study of HER-2-directed therapy, including trastuzumab, pertuzumab, or lapatinib, suggested sustained response to these agents among nine patients with HER-2 mutated gallbladder cancer. However, despite a higher incidence of HER-2 mutations in cholangiocarcinoma compared with gallbladder cancers, five patients with HER-2 mutated cholangiocarcinoma showed no response to HER-2-directed therapy in the same study [36].

Nascent prospective data are beginning to inform the use of these agents in hepatobiliary cancers. A 2021 analysis of 39 patients with HER-2 positive metastatic biliary tract cancers who were treated with trastuzumab plus pertuzumab while enrolled in the phase II My Pathway umbrella study found an ORR of 23% [37]. This finding opens the door for a randomized phase III trial examining targeted therapy for patients with HER-2-positive biliary tract cancers. Zanidatamab, an anti-HER2 bispecific antibody, demonstrated anti-tumor activity in a phase I study of 20 patients with metastatic HER2-positive biliary tract cancer whose disease progressed on prior therapy, with an ORR of 40% [38]. This agent was granted breakthrough therapy status by the FDA for this patient population and is being studied in the ongoing phase II HORIZON trial. The antibody–drug conjugate trastuzumab deruxtecan recently demonstrated an ORR of 36.4% in the phase II HERB trial among 22 patients with metastatic HER2-positive biliary tract cancer. Notably, this trial also included eight patients with HER-2 low-expressing tumors with a 12.5% ORR [39]. While there was an increased incidence of interstitial lung disease seen in the trial, including two fatal cases, these results suggest a new role for targeted therapy to improve outcomes in this patient population.

### 2.7. Immune Checkpoint Inhibitors

Immune checkpoint inhibitors (ICIs) are a group of antibody drugs that target immune checkpoint proteins, including cytotoxic T-Lymphocyte agent-4 (CTLA-4) and programmed death-1/programmed death-ligand-1 (PD-1/PDL-1.) ICIs have revolutionized the treatment of solid tumor malignancies since the first FDA approval of ipilimumab in 2011 for metastatic melanoma. Microsatellite instability (MSI) refers to regions of repeating DNA that change in length when underlying DNA repairing mismatch repair (MMR) proteins are mutated and do not function correctly [40]. In 2017, the FDA granted accelerated approval to the PD-1 inhibitor, Pembrolizumab, for tissue-agnostic tumors demonstrating MSI-H or MMR deficiency. The rate of MSI-H hepatobiliary cancers ranges from 0 to 60%, depending on the microsatellite markers used for testing [41]. MSI-H gallbladder cancers occur at increased prevalence among patients with a history of chronic cholecystitis; pancreaticobiliary maljunction is associated with a higher prevalence of MSI-H status in both gallbladder and cholangiocarcinoma.

The current standard of care for cholangiocarcinoma is a combination of chemotherapy and immune checkpoint inhibition based on the results of the TOPAZ-1 trial. This was a phase III study that evaluated the combination of gemcitabine and cisplatin combined with either the PD-1 inhibitor Durvalumab or placebo. The study demonstrated 24.9% overall survival at 24 months in the Durvalumab arm vs. 10.4% in the placebo arm.

Additional studies evaluating immune checkpoint inhibitors in hepatobiliary cancers have been explored. Pembrolizumab, a PD-1 inhibitor, has been shown to be effective in the treatment of patients with MSI-H/dMMR tumors, with 22 cholangiocarcinoma patients included in the phase II KEYNOTE-158 study [42]. In the subgroup analysis of patients with MSI-H/dMMR cholangiocarcinoma, pembrolizumab conferred an ORR of 40.9% [42]. TMB-H cholangiocarcinoma also showed a response to pembrolizumab. Pembrolizumab is approved in the second-line treatment of metastatic MSI-H, dMMR, or TMB-H solid tumors, including hepatobiliary cancers [43]. Pembrolizumab also shows promise in combination with Lenvatinib as a subsequent line therapy for patients with advanced hepatobiliary cancer not previously exposed to a checkpoint inhibitor, based on the results of the phase II LEAP-005 study [44].

Nivolumab, also a PD-1 inhibitor, showed an ORR of 22% among 46 patients with advanced hepatobiliary cancer [45]. In this study, all patients who responded to treatment with nivolumab had MMR-proficient tumors. Based on these results, nivolumab is accepted as a subsequent line treatment for patients in this population who have not been previously exposed to a checkpoint inhibitor.

### 2.8. NTRK

The family of neurotrophic tropomyosin kinase receptors (NTRK) is a transmembrane tyrosine kinase that is integral to neural development through the activation of growth signaling pathways. NTRK protein fusions have been implicated in multiple solid tumor malignancies [46]. NTRK inhibitors Intrectinib and Larotrectinib are accepted as targeted treatment options for patients with metastatic biliary tract cancers harboring NTRK fusion genes, which make up <1% of these cancers. This recommendation is based on umbrella studies of NTRK inhibitors in solid tumors [47,48].

### 2.9. VEGF

Vascular endothelial growth factor (VEGF) is a potent growth factor that stimulates angiogenesis. VEGF has been shown to promote angiogenesis in tumors, thus contributing to cancer cell growth and proliferation. VEGF targeting agents have been shown to slow growth in other solid tumors [Elebiyo]. VEGF is overexpressed in up to 75% of hepatobiliary malignancies [2]. In a phase II trial, sorafenib, given in combination with gemcitabine, improved survival compared with gemcitabine plus placebo among patients with unresectable biliary tract cancer who developed liver metastases following resection [49]. These findings are not yet supported by phase III studies. Similarly, regorafenib has been shown in phase II studies to have clinical activity and provide modest survival benefit versus placebo in patients with metastatic biliary tract cancer [50,51].

### 2.10. RET

The RET proto-oncogene is a protein receptor tyrosine kinase that, when constituently activated, can lead to the development of cancer. RET is a rare mutation, occurring in only 1.8% of all solid tumors as either a fusion, amplification, or gene mutation [52]. RET alterations have been shown to occur in only 1.6% of hepatobiliary tract cancers [53]. The only available clinical trial data on RET targeting agents in cholangiocarcinoma is from a phase II study of Selpercatinib, a RET tyrosine kinase inhibitor with highly selective binding. In the study, one patient with biliary tract cancer derived a response and remained in the study at the time of data cutoff [54]. Given the tissue agnostic approval for Selpercatanib, this would be an appropriate choice for a patient with cholangiocarcinoma harboring these genetic aberrations.

### 2.11. Future Therpatuic Targets

The treatment landscape for cholangiocarcinoma is rapidly evolving, with several novel therapeutic strategies in both pre-clinical and clinical development. Antibody–drug conjugates (ADCs) allow targeting cell surface receptors to deliver cytotoxic payloads and reduce systemic toxicities. Adoptive cellular therapies, including chimeric antibody receptor T-cell (CAR-T) and nature killer cell (NK cell) therapies, aim to harness the immune response beyond only targeting immune checkpoint proteins. The utilization of novel immunotherapies has been deployed to help overcome both primary and acquired resistance to immune checkpoint inhibitors. Additionally, drugs targeting the tumor microenvironment and tumor metabolism are under investigation for the treatment of multiple solid tumors, including those of the biliary tract.

### 2.12. Antibody–Drug Conjugates

Antibody–drug conjugates (ADCs) are novel drug delivery mechanisms with two primary components: An antibody capable of targeting proteins expressed on the cell surface as well as a ‘payload’ of a cytotoxic chemotherapeutic agent [55]. Several promising cell surface protein targets have been identified that are under pre-clinical investigation for the treatment of cholangiocarcinoma.

Yashima et al. conducted a pre-clinical study evaluating an ADC comprised of an HER-2-targeting antibody in conjunction with a trastuzumab emtansine payload in the CCA cell line. The study demonstrated that cell death was increased in HER-2 overexpressing CCA cells when compared to HER-2 negative cells by IHC GPC-1 is a cell surface proteoglycan receptor that contributes to cellular growth and proliferation by acting as a co-receptor with other pro-growth cellular receptors [55]. GPC-1 overexpression has been demonstrated in up to 47% of CCA tumor specimens [56]. ADCs targeting GPC-1 have been evaluated in pre-clinical CCA cell lines.

Yokota et al. investigated a GPC-1 targeting ADC with the drug payload of monomethyl auristatin F (MMAF), an anti-neoplastic agent that is a component of balantamb mafodotin, that is commonly employed in multiple myeloma [56]. In their study, it was found that this GPC-1 targeting ADC showed significant anti-tumor activity in both in vivo and in vitro models. Additional work demonstrated that GPC-1 knockout mice also had anti-tumor growth activity. Additional work by Zhu et al. established intra-adhesion molecule-1 (ICAM-1) as a highly expressed cell surface marker in CCA cell lines, representing a viable therapeutic target. Additional work demonstrated increased cytotoxicity when an antibody targeting ICAM-1 was conjugated with a cytotoxic payload when tested in both in vivo and in vitro experiments [57].

### 2.13. Adoptive Cellular Therapies

MUC-1 is a group of epithelial glycoproteins that have gained significant traction as potential anti-neoplastic targets, given their integral functioning in cellular growth and potentiation [58]. Lymphokine-activated killer (LAK) cells are an adoptive cellular therapy consisting of T-cells, natural killer cells (NK cells), and NK-T cells. Pre-clinical models have shown that an ADC combining a MUC-1 targeting antibody with the staphylococcal enterotoxin A (SEA) has activity against cholangiocarcinoma cell lines and that this activity can be augmented by LAK cells.

Additional studies of CAR-T cells in CCA have been conducted. One pre-clinical study evaluating a MUC-1 CAR-T cellular therapy in a CCA cell line demonstrated significant upregulation in cytokine production of IFN-γ and granzyme B when compared to un-transduced T-cells [59].

Clinical activity of cellular therapies in CCA has also been demonstrated. In one report, an EGFR-binding and CD133-restricted CAR-T combination was shown to provide 8.5 months of partial response in a patient with refractory CCA [60].

### 2.14. Cancer Vaccines

Cancer vaccines are currently under development in a multitude of solid tumors, including hepatobiliary malignancies. Several different approaches to utilizing cancer vaccines in CCA have been pursued, including mRNA-based technologies and personalized cancer vaccines. Huang et al. conducted a retrospective study that characterized the immune infiltrative potential of various antigens associated with CCA [61]. In their study, it was found that CD247, FCGR1A, and TRRAP were three antigens with expression patterns correlating with superior infiltration of T-cells and a subsequent increase in tumor inflammation of immune cells. These antigens represent promising targets for future CCA-directed vaccines.

## 3. Discussion

Cholangiocarcinoma is often diagnosed at late stages and is associated with an overall poor prognosis. Historically, these cancers have been very difficult to treat with systemic chemotherapy, and overall survival in patients with metastatic disease has been less than a year. The advent of whole exome sequencing has ushered in an era of targeted therapies for a multitude of solid tumor malignancies. Therapies targeting FGFR and IDH-1 have gained FDA approval after progression on front-line therapies for patients with cholangiocarcinoma. Additionally, less common driver mutations, such as BRAF and NTRK, have been investigated and may provide clinical benefit in a certain subset of patients. These mutations are very infrequently mutated, and prospective data on their usage in cholangiocarcinoma are limited.

The incorporation of immune checkpoint inhibitors into the treatment paradigm for cholangiocarcinoma has become the standard of care, with further studies showing clinical benefit in patients with MSI-H tumors. Despite these successes, the vast majority of patients do not derive a clinical benefit from immune checkpoint inhibitors. A significant number of patients with cholangiocarcinoma have either primary or acquired resistance to ICI therapies. Novel strategies have been explored to engage other components of the immune system to overcome this resistance. Adoptive cellular therapies, including invariant-NK cells, CAR-T, and cancer vaccines, have shown early promise; however, they are currently in the early clinical stage of development.

## Figures and Tables

**Figure 1 ijms-25-00543-f001:**
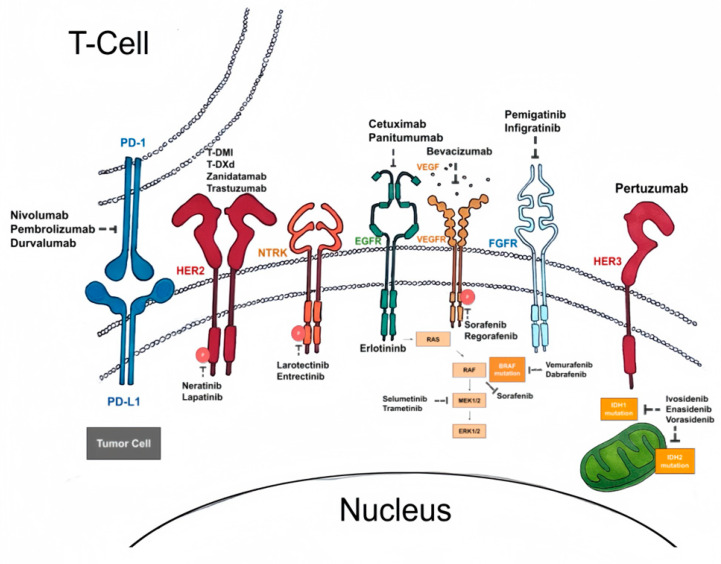
Overview of current therapeutic targets in cholangiocarcinoma.

**Table 1 ijms-25-00543-t001:** Clinical trials by targeted agent.

NTC Number	Phase	Disease	Intervention	Outcome	Status
NCT05876754	IIIb	Unresectable and/or metastatic CCA	ivosidenib (IDH1)	AEs, SAEs, QT prolonging events, ECOG, lab abnormalities, changes from baseline of labs and vitals	Enrolling
NCT03603834	II	Resectable CCA	neoadjuvant mFOLFOXIRI	ORR	Enrolling
NCT04989218	I and II	Resectable and high-risk feature ICC	durvalumab/MEDI4736 (anti-PD-L1), tremelimumab (CTLA-4), GemCis	ORR	Enrolling
NCT05348811	II	Unresectable and/or metastatic ICC	GEMOX hepatic arterial infusion chemotherapy (HAIC), donafenib (small molecule multikinase inhibitor), sintilimab (anti-PD-1)	ORR and hORR	Enrolling
NCT03820310	II	Treated with radical resection completely ICC	Autologous Tcm cellular immunotherapy and traditional therapy	PFS, 2-year survival	Enrolling
NCT02773485	III	Unresectable and nonmetastatic CCA	High-dose radiation with systemic themotherapy	OS	Enrolling
NCT03656536	III	Unresectable and/or metastatic CCA	pemigatinib (FGFR-inhibitor) vs. GemCis	PFS	Enrolling
NCT05655949	II	Unresectable and/or metastatic IHC	Y-90 SIRT with durvalumab (anti-PD-L1), GemCis	PFS, grade 3 or higher AE	Enrolling
NCT05532059	II	Unresectable and/or metastatic CCA	lenvatinib (VEGFR1-3), tislelizumab (anti-PD-1), GemCis	ORR	Enrolling
NCT05400902	II	Unresectable ICC with at least one assessable intrahepatic lesion	FOLFOX6 hepatic arterial infusion chemotherapy, sintilimab (anti-PD-1), bevacizumab (VEGF-i)	ORR	Enrolling
NCT05290116	II	Unresectable ICC with at least one assessable intrahepatic lesion	FOLFOX6 hepatic arterial infusion chemotherapy, tislelizumab (anti-PD-1), apatinib (TKI)	ORR	Enrolling
NCT05251662	II	Unresectable CCA	sintilimab (anti-PD-1), GEMOX, IBI305 (bevacizumab biosimilar)	ORR	Enrolling
NCT05430698	II	Resectable perihilar CCA with positive metastatic LNs	Adjuvant PD-1 antibody, GEMOX	12 mo relapse-free survival rate	Enrolling
NCT04295317	II	Resectable ICC	camrelizumab (anti-PD-1) and capecitabine	Recurrence-free survival	Enrolling
NCT05823311	III	Unresectable and/or metastatic CCA	lenvatinib (VEGFR1-3), tislelizumab (anti-PD-1), GemCis	ORR	Enrolling
NCT04954781	II	Unresectable or metastatic CCA	Transcatheter arterial chemoembolization (TACE) tislelizumab (anti-PD-1)	ORR	Enrolling
NCT05010668	II	Unresectable and/or metastatic ICC	Cryoablation, sintilimab (anti-PD-1), lenvatinib (VEGFR1-3)	ORR	Enrolling
NCT04299581	II	Unresectable and/or metastatic ICC	Cryoablation and Camrelizumab (Anti-PD-1)	ORR	Enrolling
NCT04891289	II	Unresectable and liver metastasis ICC	Gemcitabine, oxaliplatin, floxuridine, and dexamethasone pump	PFS	Enrolling
NCT03633773	I and II	Nonsurgical candidate, ICC	MUC-1 CART cell immunotherapy	DCR	Enrolling
NCT05557578	II	Resectable ICC and high-risk recurrence	Neoadjuvant tislelizumab (anti-PD-1) and GEMOX	ORR, R0 resection rate	Enrolling
NCT04961970	III	Unresectable or local metastatic ICC	FOLFOX hepatic arterial infusion chemotherapy vs. GemCis	OS	Enrolling
NCT04782804	I and II	Post R0 resection ICC	Adjuvant tislelizumab (anti-PD-1) and capecitabine	RFS	Enrolling
NCT05174650	II	Non-resectable ICC and FGFR2+	atezolizumab (anti-PD-L1) and derazantinib (FGFR1–3 kinase inhibitor)	ORR	Enrolling
NCT04454905	II	Unresectable and/or metastatic ICC	camrelizumab (anti-PD-1) and apatinib (TKI)	PFS	Enrolling
NCT04238637	II	Locally advanced OR limited metasized intrahepatic BTC	Y-90 SIRT with durvalumab (anti-PD-L1) or tremelimumab (CTLA-4i)	ORR	Enrolling
NCT05010681	II	Unresectable and/or metastatic ICC or HCC	sintilimab (anti-PD-1), lenvatinib (VEGFR1-3)	ORR	Enrolling
NCT05678270	II	Unresectable, recurrent or metastatic ICC and FGFR2+	gunagratinib (FGFR-inhibitor)	ORR	Enrolling
NCT05921760	I, II	Unresectable and/or metastatic CCA, IDH1+	ivosidenib, nivolumab (anti-PD1), ipilimumab (CTLA-4i)	Dose-limiting toxicities (DLT), recommended combination dose (RDC), ORR, AEs, SAEs	Enrolling
NCT04353375	II	Unresectable and/or metastatic ICC, FGFR2+	HMPL-453 tartrate (FGFR1, FGFR2, FGFR3 antagonist)	ORR	Enrolling
NCT05835245	II	Unresectable and/or metastatic ICC	Cryoablation with sintilimab (anti-PD-1), lenvatinib (VEGFR1-3)	ORR	Enrolling
NCT05978609	II	Unresectable and/or metastatic CCA	Candonilimab (PD-1/CTLA-4), GemCis	ORR	Enrolling
NCT04951141	I	Unresectable liver tumor, GPC3+	anti-GPC3 CAR-T cells	AEs	Enrolling
NCT04708067	I	Unresectable and/or metastatic ICC	bintrafusp alfa and hypofractionated radiation therapy	AEs	Enrolling
NCT05672537	II	Resectable ICC	Neoadjuvant durvalumab (anti-PD-L1), surgery, GemCis	RFS	Enrolling
NCT05239169	II	Unresectable and/or metastatic BTC	durvalumab (anti-PD-L1), tremelimumab (CTLA-4), capecitabine	RFS	Enrolling
NCT05223816	IIa/IIb	Unresectable and/or metastatic ICC	VG-161 (IL-12, IL-15, PD-L1), nivolumab (anti-PD1)	OS, ORR, PFS	Enrolling
NCT05220722	I, II	Unresectable and/or metastatic ICC	SD-101 (TLR 9 agonist), pembrolizumab (anti-PD-1), nivolumab (anti-PD1), ipilimumab (CTLA-4i)	Safety, maximum tolerable dose (MTD), ORR	Enrolling
NCT06037980	II, III	Resectable BTC at high risk for recurrence	GemCis, nabpaclitaxel vs. upfront surgery	PFS	Enrolling
NCT03779035	III	BTC after curative resection	Adjuvant GemCis vs. capecitabine	DFS	Enrolling
NCT04634058	II	Unresectable and metastatic ICC	PD-L1 and CTLA-4	ORR	Enrolling
NCT03364530	II	Nonmetastatic unresectable ICC	Hepatic intra-aterial Gemcitabine-Oxaliplatin	ORR	Enrolling
NCT04298021	II	Unresectable and/or recurrent BTC	AZD6738 (ATR Kinase-i), durvalumab (anti-PD-L1), olaparib (PARP-i)	DCR	Enrolling
NCT04298008	II	Unresectable and/or recurrent BTC	AZD6738 (ATR Kinase-i) and durvalumab (anti-PD-L1)	DCR	Enrolling
NCT05727176	II	Unresectable and/or metastatic CCA, FGFR2+	Futibatinib (FGFR1–4 inhibitor)	ORR	Enrolling
NCT06081322	I	Advanced cholangiocarcinoma	177Lu-EB-FAPI PRRT (FAP inhibitor molecule)	Hematoxicity, ORR, hepatotoxicity, renal toxicity	Enrolling
NCT04413734	II	Unresectable and/or metastatic BTC	triprilumab (anti-PD-1), GemCis	PFS	Enrolling
NCT04251715	IId	Unresectable ICC	mFOLFIRINOX, hepatic arterial infusion of floxuridine, dexamethasone, and systemic mFOLFIRI	Incidence of abnormal liver function, DCR	Enrolling
NCT05568680	I	Recurrent or relapsed advanced mesothelial CCA	SynKIR-110 (anti-mesothelin)	Safety and feasibility	Enrolling
NCT04565275	I, II	Unresectable and/or metastatic CCA, FGFR+	ICP-192 (pan-FGFR inhibitor)	AEs, MTD, OBD, RP2D, ORR	Enrolling
NCT03991832	II	Adenocarcinoma BTC, IDH+	olaparib (PARP-i), durvalumab (anti-PD-L1)	ORR, DCR	Enrolling
NCT04264260	II	ICC	colchicine	OS	Enrolling
NCT04669496	II, III	Resectable ICC and high-risk recurrence	neoadjuvant toripalimab (anti-PD-1), lenvatinib (VEGF1-3), GEMOX	Event free survival	Enrolling
NCT05286814	II	Unresectable and/or metastatic CCA	M9241 (IL-12 heterodimers) hepatic artery infusion pump and systemic chemotherapy, dexamethasone	ORR	Enrolling
NCT03801083	II	Unresectable, recurrent, or metastatic BTC	Autologous tumor-infiltrating lymphocytes (TIL)	ORR	Enrolling
NCT05242822	I	Advanced stage CCA, FGFR2+ and/or FGFR3+	KIN-3248 (small molecule pan-FGFRi)	DLT, AEs, ORR, DCR, duration of response, PFS	Enrolling
NCT05724563	II	Unresectable and/or metastatic BTC	domvanalimab (anti-TIGIT), zimberelimab (anti-PD-1)	ORR	Enrolling
NCT05913661	II	Unresectable and/or metastatic ICC	pemigatinib (FGFR-inhibitor) and anti-PD-1	ORR	Enrolling
NCT04645160	I, II	Unresectable BTC	tivozanib (VEGF-i)	ORR, RP2D	Enrolling
NCT04068194	I, II	Unresectable and/or metastatic ICC	peposertib (DNA-PK-i), avelumab (anti-PD-L1), RT	MTD, ORR	Enrolling
NCT05565794	II	curatively treatable localized ICC, FGFR2+	pemigatinib (FGFR-inhibitor)	ORR	Enrolling
NCT04526106	I, II	Unresectable and/or metastatic CCA	RLY-4008 (FGFR2-i)	MTD, AEs, SAEs, ORR	Enrolling
NCT05620498	II	Potentially resectable ICC and GBC	tislelizumab (anti-PD-1), lenvatinib lenvatinib (VEGFR1-3), GEMOX	ORR	Enrolling
NCT05506943	II, III	Unresectable, metastatic, or recurrent BTC	CTX-009 (Delta-like ligand 4/Notch-1 (DLL4) and VEGF-A inhibitor)	BOR	Enrolling
NCT03942328	I, II	Unresectable ICC	intra-tumoral injection of autologous dendritic cells and prevnar vaccine after EBRT, atezolizumab (anti-PD-L1), bevacizumab (VEGF-i)	incidence of significant toxicity, PFS	Enrolling
NCT05564403	II	Unresectable or recurrent BTC, RAS/RAF/MEK/ERK+	binimetinib (MEK-i), FOLFOX	OS	Enrolling
NCT05211323	II	Unresectable or metastatic cHCC-CC	atezolizumab (anti-PD-L1), bevacizumab (VEGF-i), GemCis	PFS	Enrolling
NCT03212274	II	BTC, IDH1/2+	olaparib (PARP-i),	ORR	Enrolling
NCT05805956	I, II	Unresectable and/or metastatic BTC, HER2+	IMM2902 (anti-CD47/anti-HER2 bispecific antibody)	MTD, RP2D, AEs, ORR, DOR, DCR, PFS	Enrolling
NCT05411133	I	Unresectable and/or metastatic BTC, CDH17+	ARB202 (CDH17-CD3 bispecific T-cell engager antibody)	SAEs	Enrolling
NCT04801095	I	Unresectable and/or metastatic BTC	WM-S1-030 (inhibitor for mtRTK)	DLT, AEs, SAEs	Enrolling
NCT04430738	I, II	Unresectable and/or metastatic BTC	tucatinib (HER2 tyrosine kinase inhibitor), trastazumab (anti-EGFR), oxaliplatin-based chemotherapy, pembrolizumab (anti-PD-1)	DLT; AEs; incidence of lab abnormalities, DLTs, dose alterations	Enrolling
NCT05285358	I	BTC with peritoneal metastasis	pressurized intraperitoneal aerosolized nab-paclitaxel, GemCis	AEs	Enrolling
NCT06010862	I	Unresectable and/or metastatic BTC, CEA+	CEA-targeted CAR-T	AEs, MTD	Enrolling
NCT05185947	II	BTC with peritoneal metastasis	intravenous and intraperitoneal paclitaxel, nilotinib (transduction inhibitor of BCR-ABL, c-kit and PDGF)	OS, PFS, safety, tolerability, QOL, ORR	Enrolling
NCT05991518	I, II	Unresectable and/or metastatic BTC, HER2+	IAH0968 (afucosylated anti-EGFR2), GemCIs	AE, SAEs, DLT, ORR	Enrolling
NCT06126406	I	Unresectable and/or metastatic BTC, CEA+	CEA-targeted CAR-T	AEs, MTD	Enrolling
NCT03907852	I, II	BTC, MSLN+	Gavocabtagene Autoleucel (anti-Mesothelin), fludarabine, cyclophosphamide, nivolumab (anti-PD1), ipilimumab (CTLA-4i)	RP2D, ORR, DCR	Enrolling
NCT06043466	I	Unresectable and/or metastatic BTC, CEA+	CEA-targeted CAR-T	DLT, MTD	Enrolling
NCT05277766	I	BTC with peritoneal metastasis	intraperitoneal aerosolized nano liposomal irinotecan	MTD	Enrolling

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
