# Peer review of "Current and Emerging Therapeutic Targets for the Treatment of Cholangiocarcinoma: An Updated Review"

_ijms, 2023, doi:10.3390/ijms25010543_

Round 1
Reviewer 1 Report
Comments and Suggestions for Authors
This is a good summary of molecular targets in the treatment of biliary tract malignancies, including most recent updates. It can be further improved before publication in regard to following points.
1. The manuscript frequently uses terms “hepatobiliary malignancy” and “hepatobiliary cancers”, while states “hepatobiliary malignancies comprise gallbladder cancers, ampullary carcinomas, and intrahepatic and extrahepatic cholangiocarcinomas.” However, it is widely accepted that hepatobiliary cancers include hepatocellular carcinoma, gallbladder cancer, and cholangiocarcinoma (PMID: 34030131, PMID: 32045030). It’s not clear whether the current manuscript address cholangiocarcinomas only, or cholangiocarcinoma plus hepatocellular carcinoma. Authors may want to clarify this.
2. The main body of this manuscript looks quite random: a variety of RTK inhibitors are interspace with other categories (metabolic, ICIs). Is this a special consideration to this?
3. Authors have listed only four clinical trials, which is apparently less than registered trials at clinicaltrials.gov and other recent reviews (e.g. PMID: 30743998). Please describe any criteria to include these trials (or not include other trials).
4. (Line 23) 12,000 cases are most likely “Gallbladder & other biliary cancers” excluding intrahepatic bile duct. Thus, this piece of information can be misleading when it’s combined with “biliary tract cancers”. In addition, authors can cite the most recent Cancer Statistics published in 2023 (PMID: 36633525).
5. (Line 215) “The current stand of care of cholangiocarcinoma is a combination of chemotherapy and immune checkpoint inhibition”. Is there any reference, like guidelines supporting this?
6. Figure 1 requires a descriptive legend, and should be referred to in the main text.
Author Response
- The manuscript frequently uses terms “hepatobiliary malignancy” and “hepatobiliary cancers”, while states “hepatobiliary malignancies comprise gallbladder cancers, ampullary carcinomas, and intrahepatic and extrahepatic cholangiocarcinomas.” However, it is widely accepted that hepatobiliary cancers include hepatocellular carcinoma, gallbladder cancer, and cholangiocarcinoma (PMID: 34030131, PMID: 32045030). It’s not clear whether the current manuscript address cholangiocarcinomas only, or cholangiocarcinoma plus hepatocellular carcinoma. Authors may want to clarify this. The paper has been narrowed to discuss cholangiocarcinoma and malignancies of the bile tract.
- The main body of this manuscript looks quite random: a variety of RTK inhibitors are interspace with other categories (metabolic, ICIs). Is this a special consideration to this? The whole paper has been edited to flow more easily.
- Authors have listed only four clinical trials, which is apparently less than registered trials at clinicaltrials.gov and other recent reviews (e.g. PMID: 30743998). Please describe any criteria to include these trials (or not include other trials). An updated list of trials has been added.
- (Line 23) 12,000 cases are most likely “Gallbladder & other biliary cancers” excluding intrahepatic bile duct. Thus, this piece of information can be misleading when it’s combined with “biliary tract cancers”. In addition, authors can cite the most recent Cancer Statistics published in 2023 (PMID: 36633525). Updated references has been added and this has been clarified.
- (Line 215) “The current stand of care of cholangiocarcinoma is a combination of chemotherapy and immune checkpoint inhibition”. Is there any reference, like guidelines supporting this? The reference for the corresponding clinical trials has been added.
- Figure 1 requires a descriptive legend, and should be referred to in the main text. This has been added.
Reviewer 2 Report
Comments and Suggestions for Authors
Authors are requested to revise the manuscript according to the suggestions.

Language is fine, few typographical mistakes should be rectified.
Author Response
The manuscript has been edited to include a title that more reflects the information covered in the review which is focused on cholangiocarcionma and bile duct malignancies. The flow, organization, and order of the sections as well as the references have all been updated. Additionally, the minor edits have been addressed.
Round 2
Reviewer 1 Report
Comments and Suggestions for Authors
The revised manuscript has been improved in terms of scope and summary of recent clinical progress. However, it's still less structured regarding the category of molecular targets. A similar review was published on Medicina, another MDPI journal in 2019 (doi:10.3390/medicina55020042). I appreciate that the manuscript has included more recent studies and tried not to follow the same structure. But it could be more well organized.
Table 1 can include both ongoing and recently completed trials (or in separate tables)
Please also proofread the manuscript for typos. (e.g. "nucleas" should be "nucleus" in Figure 1)
Author Response
The manuscript has been edited for any grammatical errors. Additional studies on ADCs in CCA have also been added.
Reviewer 2 Report
Comments and Suggestions for Authors
Authors have revised the manuscript well and all the suggestions recommended were included.
Author Response
Thank you
Round 3
Reviewer 1 Report
Comments and Suggestions for Authors
Manuscript has been improved.
Minor points:
1) Please check typos. e.g. "FGRF" in line 13, "tyrosine receptors" in line 117, "receptor tyrosine-protein kinase recrptor" in line 177.
2) Please keep citation format consistent. Current citations in lines 110 [Phillip], 253 [Elebiyo] and 257 [Moehler] are different from others.
3) Please keep acronyms consistent acorss the manuscript, e.g. in section 2.6 and 2.12 where "HER-2", "HER2" and "Her-2" appear.
Comments on the Quality of English LanguagePlease proofread the manuscript. See above comments.
Author Response
Thank you very much for your review of our manuscript. We've made the following corrections:
- The paper was reviewed and minor grammatical errors were corrected
- Abbreviations were harmonized throughout the paper
- References were corrected.